# Processability and Separability of Commercial Anti-Corrosion Coatings Produced by In Situ Hydrogen-Processing of Magnetic Scrap (HPMS) Recycling of NdFeB

**DOI:** 10.3390/ma17112487

**Published:** 2024-05-21

**Authors:** Laura Grau, Peter Fleissner, Spomenka Kobe, Carlo Burkhardt

**Affiliations:** 1Jožef Stefan International Postgraduate School, Jamova Cesta 39, 1000 Ljubljana, Slovenia; 2Institute of Precious and Technology Metals (STI), Pforzheim University, Tiefenbronner Straße 65, 75175 Pforzheim, Germany; 3Department for Nanostructured Materials (K7), Jožef Stefan Institute, Jamova Cesta 39, 1000 Ljubljana, Slovenia

**Keywords:** HPMS, NdFeB recycling, coatings, IPMSM, design for recycling

## Abstract

The recycling of NdFeB magnets is necessary to ensure a reliable and ethical supply of rare earth elements as critical raw materials. This has been recognized internationally, prompting the implementation of large-scale legislative measured aimed at its resolution; for example, an ambitious recycling quote has been established in the Critical Raw Materials Act Successful recycling in sufficient quantities is challenged by product designs that do not allow the extraction and recycling of these high-performance permanent magnets without excessive effort and cost. This is particularly true for smaller motors using NdFeB magnets. Therefore, methods of recycling such arrangements with little or no dismantling are being researched. They are tested for the hydrogen-processing of magnetic scrap (HPMS) method, a short-loop mechanical recycling process. As contamination of the recycled material with residues of anti-corrosion coatings, adhesives, etc., may lead to downcycling, the separability of such residues from bulk magnets and magnet powder is explored. It is found that the hydrogen permeability, expansion volume, and the chosen coating affect the viable preparation and separation methods as recyclability-relevant design features.

## 1. Introduction

The benchmark of having to recycle 25% of NdFeB by 2030 was set by the Critical Raw Materials Act (CRMA) [1], recognizing the significant gap between the projected supply and demand of NdFeB magnets due to their increasing use in high-power permanent magnet applications such as electric vehicles or wind turbines [2]. Rare earth deposits are scarce and difficult to mine: primary mining and chemical processing pose a threat to workers and the environment [3]. Recycling could break the vicious circle of supplying environmentally critical materials such as NdFeB to the clean energy transition and electromobility. Hydrogen-processing of magnetic scrap (HPMS) for sintered NdFeB enables short-loop recycling, avoiding multiple mechanical and chemical refining processes [4]. By using hydrogen to embrittle and pulverize the magnets, recycling can take place from the powder stage, where the powder is prepared for a subsequent shaping and sintering process. This is energetically favorable compared to hydro- and pyrometallurgical recycling [5].

Due to their high reactivity, NdFeB permanent magnets must be protected from corrosion during operation. Corrosion protection is usually achieved through the use of a suitable coating. However, the production of high-performance magnets via HPMS requires the high purity of the processed recyclate powders, with the need for a low oxygen and carbon content, as well as a low content of other unwanted chemical elements, which may be present in the form of coatings, adhesives and other residues, thus affecting the quality of the recycled magnet. If detrimental, such residues must be separated from the NdFeB bulk either before or after the HPMS process.

For large applications such as wind turbines, electric vehicle traction motors or computer hard disk drives, automated dismantling and extraction of the NdFeB permanent magnets is suggested for commercial viability [6,7]. This process is hereby referred to as ex situ, as the magnets are removed prior to the HPMS, allowing for the removal of the coatings and adhesives, which is advantageous to achieve non-contaminated NdFeB powder. However, other applications, particularly small motors, which may not be worth dismantling due to the small amount of material involved but which occur in large volumes, are also important sources of scrap. In order to present a solution for the recovery of these important scrap sources for the circular economy, the processing of coated and passivated NdFeB in a non-dismantled state (in situ) is being investigated.

## 2. Materials and Methods

In this section, the used general methods (Section 2.1), the test subjects and exact test setups (Section 2.2) are explained, followed by a short review of commonly used coating materials (Section 2.3).

### 2.1. Methods

Hydrogen-processing of magnetic scrap (HPMS) is a patented process developed by researchers at the University of Birmingham [8]. Magnets are placed in a reactor, which is then evacuated and filled with pure hydrogen at adjustable pressures at laboratory temperature. The Nd in the grain boundaries reacts with hydrogen to form NdH_~2.7_ and the hard-magnetic Nd_2_Fe_14_B matrix phase accommodates a number of interstitial hydrides, expanding by up to 5% in volume [9]. This process is illustrated in Figure 1, together with a common subsequent processing route by pressing and sintering. The decomposition of the Nd-rich phase reduces the integrity of the matrix, leading to embrittlement, and the volume expansions cause the grains to be pushed apart, resulting in coarse, friable aggregates.

The material is kept in a bespoke reactor in a hydrogen atmosphere until the hydrogen consumption is reduced to almost zero, assuming the reaction has reached equilibrium. The powder is then partially degassed by evacuation and the powder is removed from the reactor. The result after the HPMS can range from a single crystal powder with particle sizes below 10 µm to aggregates several mm in size. This powder has to be milled to counteract the grain growth during sintering. To initiate this process, the hydrogen must have a pathway to enter the microstructure at the grain boundaries. Dense coatings must therefore be removed or partially fractured. If the coating has not been completely removed prior to the HPMS, it may disintegrate, remain as mostly uniform carcasses or flakes, or adhere to the individual pieces of NdFeB [10], and the residues are typically separated by a manual post-treatment such as sieving or sifting.

If in situ HPMS is performed on a non-dismantled array, a hydrogen pathway may be provided by damage to the coating. However, the magnet may not disintegrate properly if there is insufficient space for expansion [11].

### 2.2. Experimental

A range of end-of-life and production scrap rotors less than 100 mm in diameter were used in this study. As there were no end-of-life rotors available with an Ni-based coating, only relatively fresh production scrap, hard disk drive magnets with different Ni and Ni-Cu coatings were added, as the behavior of the material can vary greatly depending on the corrosion state of the NdFeB bulk, and these supplementary magnets were necessary to obtain representative results in the powder separation experiments. An overview of the samples is provided in Table 1. Several internal permanent magnet synchronous motor (IPMSM) rotors as well as surface-mounted permanent magnet synchronous motor (SPMSM) rotors with an outer rotor (OR) or inner rotor (IR) arrangement were used. The rotor types and the test method are shown in Figure 2. The coatings were identified and characterized by SEM-EDS. A short review of the relevant characteristics and variations of the common anti-corrosion coatings relevant to this study is provided in Section 2.3 Materials to support the later discussion.

As a baseline for the in situ testing, rotors with passivated magnets were tested without any preparation. They were exposed to hydrogen to determine if the passivation was permeable. Tests on rotors with metal-coated magnets or magnets with passivations that were found to be non-permeable to hydrogen in previous tests were carried out after damaging the coating by scoring or sandblasting. For the HPMS-processing trials, a reactor containing rotors with attached magnets was filled with hydrogen at 3 bar. When the pressure dropped below 2 bar due to the ongoing reaction, the reactor was refilled until the pressure drop per hour was less than 0.1 bar. If the magnets did not disintegrate due to a lack of expansion volume, another attempt was made with the bridges cut once, allowing the bridge to deform and thus to accommodate the expanding magnetic material, as shown in Figure 2. If this method also failed to induce a reaction, the bridge was cut on the other side as well. As a result, it fell off when the magnet was decomposed and demagnetized. Additional tests were carried out with variations in the HPMS-processing parameters to increase the particle size of the aggregates to avoid clogging by stuck particles.

The resulting powders, as well as additional HPMS powders from the single magnet samples 6 and 7 with metal coating remnants, were further tested in separation experiments by milling, followed by sieving, in a further development of an earlier method reported in Burkhardt et al. [12]. Alternatively, chemical separation was investigated by exposing the powder to an acidic liquid. This method is of secondary consideration and is further worked on by other members of the research group. ICP-OES was used to monitor the effectiveness of the separation. ICP samples are digested in aqua regia solution with 5% nitric acid using an MARS6 microwave, heating the digestive solution to 200 °C. The solution was then diluted and measured in the iCAP 7400 ICP-OES against a custom tuning solution.

Laboratory-scale vibratory ball-milling was carried out using a RETSCH MM500 nano with 3 10 min cycles at 35 Hz with a steel ball to powder ratio of 1.6:1. A Haver-Bröcker EML 200 sieve cascade with 1000 µm, 500 µm, 250 µm, 125 µm, 63 µm, 45 µm and 25 µm sieves was used for the sieving trials and jet milling with an integrated hydrocyclone stage was tested by an external supplier on a few Ni/Cu/Ni-coated magnets as a method of coating separation.

### 2.3. Materials

Anti-corrosion coatings are used due to NdFeB magnets being susceptible to corrosion, which is due to the corrosion potential of the bimetallic nature of the material, leading to intergranular corrosion through the formation of corrosion batteries [13,14]. If the highly reactive Nd in the Nd-rich phase is in an ionic bond, e.g., by oxidation, it cannot be hydrided and embrittled, nor can it act as a liquid phase during sintering. The corrosion behavior of the material itself can be improved by alloying the NdFeB system with suitable metals to reduce the highly negative standard electrode potential of the Nd-rich phase. Alloying with elements such as Al, Co, Cr, Ga, Nb, Pb, P, Sn, Ti, Zr, Ho or Ce can improve the corrosion resistance of the magnets [15,16,17,18,19,20,21]. Common metallic coatings are Zn [22,23,24], Ni [25,26,27], Ni/Cu/Ni [28,29], or less commonly, Sn [10], Al [30,31,32] and Zn-Ni [33]. Metallic coatings are often deposited by electroplating [24,26]. Other methods of depositing metallic coatings include sputtering [34], vacuum evaporation [31] or electroless plating [25]. Also common are various types of passivation [10,35,36] and organic coatings and adhesives such as epoxy resins [37,38] and composite coatings such as epoxy resins filled with ceramic and metallic particles [39,40] or even multi-layer coatings using a combination of passivation, metallic coatings and epoxy coatings.

Ni-based coatings, including Ni-Cu composites, can be produced by a variety of methods with varying benefits, but they are generally considered to provide superior corrosion protection due to their high heat resistance, durability and impermeability. Ni and Cu coatings are cathodic due to their higher standard electrode potential. They are therefore less susceptible to corrosion than the various phases of NdFeB and cannot be used as a sacrificial anode for the highly reactive neodymium-rich phase. As a result, the requirement for the coating density is high. Coatings are often multi-layered with Cu and various types of Ni alloys. High-P (high-phosphorus) Ni coatings have excellent mechanical and anticorrosion properties compared to regular Ni coatings due to their high density and amorphous structure [25,26], but they also have a porous structure [33]. Not exclusive to high-P Ni, but very common, are problems with poor adhesion and bloating of the coating. This is caused by the immersion of the magnet in electrolytes, where the grain boundaries are infiltrated and the resulting corrosion produces hydrogen under the coating, causing the coating to bloat and break [33]. Ni/Cu/Ni is a common nickel-based multilayer coating often observed on end-of-life magnets and provides excellent corrosion protection [29]. SEM images of some Ni-based coatings are shown in Figure 3.

Zinc-based coatings offer the advantage of being able to act as a sacrificial anode due to having a lower standard electrode potential than Nd, but they have the disadvantage of not being durable at higher temperatures [24,26]. Zinc coatings therefore do not need to be fully dense and can be relatively thin (about 10 µm [13]), while still providing good corrosion resistance to various chemically abrasive environments at lower temperatures due to their anodic nature [37]. Zinc coatings are inexpensive and add only small amounts of non-magnetic volume due to their low thickness [41]. Similar to Ni-based coatings, Zn-P is also an option, with higher corrosion resistance and excellent properties, to facilitate adhesion between NdFeB and, e.g., Ni-P or epoxy [33].

Passivations are suitable for well-protected designs, low-erosion environments, or under-coatings to protect the magnet before or during the application of the main coating. NdFeB can be surface passivated by exposing the magnet to an acidic liquid [33]. Different types of passivation have been observed [10], which still need to be categorized and analyzed in terms of their recyclability. The clear identification of passivations requires more advanced surface analysis methods, such as XPS, so only magnets with known passivations can be considered. Unlike the coatings mentioned above, some passivations are permeable to hydrogen, which can be advantageous for hydrogen treatments to recycle the magnet. Phosphating is a very popular treatment due to its improved corrosion resistance compared to other passivations [33], but it was found by Burkhardt et al. [10] that phosphatings can be impermeable to hydrogen, which has been confirmed several times by processing trials since the end of that study.

## 3. Results

### 3.1. In Situ Processing Trials

The sample rotors and magnets were tested with different combinations of methods, starting with the least effort and then gradually adding preparation steps to successfully produce clean NdFeB powder from the sample, as described in Section 3.2.

IPMSM: Passivated and Zn-coated magnets mounted on IPMSM rotors were tested for their processability by in situ HPMS. The rotors were not demagnetized beforehand. The phosphate-passivated magnets of sample 3 showed in the first attempt (no preparation) that the coating was not permeable to hydrogen, confirming the observations of Burkhardt et al. [12]. The Zn coatings were also impermeable. In a second attempt, the externally visible ends of the magnets were damaged to allow hydrogen access. It was observed that the lack of expansion volume can cause the coarse HPMS powder to become stuck in the steel sheet stacks. In one rotor (sample 1), it was possible to remove the powder by hitting the rotor with a mortar, but in the other cases, this was not practical. In a further test, the magnet pockets were cut as described in Section 3.2. In one case, involving a relatively large magnet (sample 2), a single cut was sufficient to cause slight deformation, with the magnet powder being loose after the treatment. In other cases (sample 3, including several similar end-of-life rotors), cutting on both sides was the only viable option to separate the magnet powder from the stacks. As an alternative to cutting the stacks, attempts were made to vary the hydrogen pressure and treatment time, with lower hydrogen pressures resulting in finer powder (see Figure 4A). A treatment at the standard pressure used of 3 bar for 1 h resulted in a Dv50 of 330 µm and a Dv90 of 632 µm, while at 1 bar and 1 h, the Dv50 value reached 280 µm and the Dv90 value reached 546 µm. The particle size distribution was measured with a Malvern Panalytical Mastersizer 3000 (Almelo, The Netherlands) with an Aero-S dispersion unit at 4 bar, using the Mie theory and a slit size of 1.25 mm for the measurement.

SPMSM: For SPMSM rotors, it was assumed that the exposed magnet surfaces would need to be sandblasted or otherwise damaged to allow the NdFeB material to disintegrate and fall off, leaving the coating backs and adhesives behind. This was not the case, as large sections of the coating backs came off, some with the adhesive still visibly attached. This was even the case for a type of magnet with a thick Ni-Cu-Ni coating with low adhesion, which can be easily peeled off by hand (see sample 6, as shown in Figure 4B). A simple method of dismantling is thermal demagnetization, which breaks down the adhesive. This should be followed by either cleaning, if the coating residue can be removed from the powder, or removing the coating altogether.

Creation of hydrogen access: In configurations where soft magnetic swarf or powder material is generated by the hydrogen access method, e.g., by sandblasting to remove Ni-based coatings or by rotary cutting of steel stacks or the coating surface, it has been found that the generated material adheres to the still magnetized NdFeB permanent magnets. As the removal of the soft magnetic swarf or powder material is very challenging, such means of providing hydrogen access are not recommended in this case due to the risk of contaminating the HPMS powder. The presence of soft magnetic phases such as α-Fe in a sintered NdFeB magnet is known to be highly detrimental to the magnetic properties [42,43].

### 3.2. Coating Separation Trials

Coating separation of Ni-based coatings: Mechanical removal of coating residues from the HPMS powder was tested on previously removed magnets and on magnet powder produced by in situ treatment. Sieving to separate the coating flakes and NdFeB material after the HPMS proved to be challenging, as the flakes and HPMS powder overlapped significantly in terms of the particle size, resulting in a low yield. Relying on the high ductility of the Ni and Cu coatings, an attempt was made to mill the HPMS powder with all the coating residues still present. The results of these experiments, both conducted on mixed production scrap and mixed end-of-life scrap, are seen in Figure 5. For lightly corroded production scrap, excellent separation was achieved in the subsequent sieving process, with Ni-Cu-Ni flakes ending up mostly in the 125 µm sieve, with small amounts in the fraction above 63 µm, and milled NdFeB powder without Ni and Cu residues below 25 µm. For highly oxidized end-of-life scrap, the separation after milling was less successful. As expected, advanced corrosion interfered with the process, resulting in non-hydrided pieces and therefore large powder agglomerates, some of which could not be completely broken down during milling. Here, the sieve fractions between 25 µm and 125 µm contained larger quantities of NdFeB as well as low levels of Ni, which was thought to be due to a brittle, cellular high phosphorous Ni layer found only in the magnets used in this particular batch of powder. Sandblasting worked for most Ni-based coatings tested, with one exception (sample 6/7 (demounted)) where the coating was deformed rather than removed (see Figure 4C), which was due to the high ductility of the material and its uncharacteristically low adhesion to the magnet.

Coating separation of Zn-based coatings: For the Zn coating residues, separation by sieving before and after milling was not possible. After sieving, most of the Zn ended up in the main NdFeB fraction, with a particle size below 25 µm. The Zn content of the powder was successfully reduced to a level not measurable by ICP by immersing the powder in a 1 molar citric acid bath for 30 min at room temperature. As an alternative, sandblasting has been shown to be a fast and reliable method of removing Zn coatings. The material losses are minimal in sandblasting. Sandblasting Zn-coated magnets with a coating thickness of an average of 18 µm resulted in an average total decrease in thickness of the sample of 24 µm per side.

## 4. Discussion

In order to assess the suitability of a coating for recycling by the HPMS, several issues need to be considered. Firstly, a magnetic coating is selected to provide optimum protection in the operating environment to ensure the desired function over the lifetime of the magnet-containing component. Good corrosion protection is also beneficial for HPMS recycling, as the high oxygen uptake in EOL magnets leads to Nd oxides that prevent proper wetting of the grain boundaries during re-sintering of the HPMS powders. This impairs the decoupling of the grains, reduces coercivity and affects the density and thus the remanence in recycled magnets [39,44]. Secondly, easy removal of the magnets from the array and good separability of the coating from the magnetic material are key to successful recycling.

Under certain conditions it is possible to treat small IPMSM rotors in situ without disassembly.

Since the HPMS powder becomes stuck in the stacks as a result of the volume expansion, only a few rotors could be processed without further mechanical measures by simply creating hydrogen access. Reliable processing is only possible by either cutting into the bridge over the magnet on both sides and allowing the bridge to fall off, or by using low hydrogen pressure and omitting the cutting step. There may be limitations if the material is highly oxidized. It is believed that tumbling during the HPMS would also be useful to clear the assembly of already degraded powder, but this has not been possible with the existing laboratory-scale HPMS reactor configuration. For SPMSM rotors, it was found that disassembly could be easily achieved by thermally demagnetizing the rotors so that the adhesive dissolves and the magnets fall off, allowing the rotors to be treated in a disassembled state.

Ni-based coatings, including Ni-Cu-Ni, provide good and also tailorable chemical and mechanical protection, which relies on both the layering of the coating as well as the chemical properties like the phosphate content. Whilst sandblasting worked for almost every magnet tested, it may not be entirely reliable for Ni-based coatings if their adhesion to the magnet is inadequate, allowing them to deform rather than be removed. Separation of Ni-based coatings by sieving may be the preferred method for most small motor applications, in which case sieving the ground HPMS powder is usually less laborious than extracting the magnets combined with individual sandblasting. Nonetheless, some losses in terms of the NdFeB material yield must be expected due to it sticking to the coating carcasses before milling (see Figure 4B) as well as after milling (see Figure 5, fractions >125 µm). Difficulties may be encountered with unsuitable (but rarely used) high-P Ni coatings, which display a brittle, cellular structure, causing them to mill finely, or a high degree of oxidation of the HPMS powder, so that the separation is not as clear. As a result, high amounts of NdFeB may be present in the sieve fractions between 25 µm and 125 µm, possibly containing some Ni contamination. In these cases, a decision must be made as to whether the contamination is tolerable for blending with cleaner material or whether the powder may be unusable when obtained by this route due to oxidation. This may impact the yield of recyclable powder produced.

For Zn-coated magnets, disassembly and mechanical removal of the coating by sandblasting or grinding should be preferred. With the investigated specimen, mechanical powder separation was not possible. Chemical removal with acids is possible, as Zn is more reactive than NdFeB, but a mechanical method that does not produce liquid waste or rely on wet chemical processes and subsequent drying is preferred to avoid waste and excessive effort. Theoretically, small amounts of Zn may be tolerable in the recycled material, as Zn evaporates during the sintering of recycled NdFeB magnets before the densification temperature is reached. The beneficial effects on Zn addition have been found by adding Zn via single-stage hot working [45], which is an entirely different procedure with mechanisms that cannot be compared to a possible Zn integration via sintering. However, larger amounts can damage the equipment or cause pores and fractures in the magnets due to the high vapor pressure.

For passivations whose low residual content due to their low thickness is not expected to significantly affect the magnetic properties of the re-sintered magnet, hydrogen permeability and good corrosion protection are most important features for successful HPMS recycling. Phosphate passivations provide the best corrosion protection but are impermeable to hydrogen. In practice, the type of passivation is often unclear unless specified or fully analyzed. Therefore, such magnets should be assumed to be hydrogen impermeable and the appropriate recycling treatments described above should be used. To increase recycling rates, a specific labeling, as foreseen in the Critical Raw Materials Act [1], is considered mandatory.

## 5. Conclusions

The recyclability of small motors with different coatings in various arrangements was evaluated and compared. No one-size-fits-all solution can provide good-quality powder at low costs for every design.

If the disassembly of small NdFeB magnet assemblies is not possible for technical or economic reasons, in situ treatment at low hydrogen pressures or by cutting the steel bridges is recommended. For passivated magnets, damage to the coating is sufficient, while Ni-based coatings (with the exception of a few high-P coatings) can be easily separated from the HPMS powder by milling and sieving, and Zn can be chemically removed, but in the latter case dismantling is the preferred option. In some assemblies, such as SPMSMs, dismantling of the magnets by thermal treatment can be very straightforward, making it ideal for automation.

The effects of the thermal treatment on the interaction of the contaminant (particularly Zn and epoxy) and the magnet should be the subject of further research. The tolerable level of contamination for the possible contaminants should be tested in further studies, as perfect separation is both technically and economically unrealistic. Based on this, it should be investigated what yield of recyclable NdFeB powder of both highly contaminated and barely contaminated samples with Ni-based coatings could be achieved with the presented method of milling and sieving. The milling process should be optimized for an industrial-scale method like jet milling to avoid material losses, which could be the subject of further investigation. The recycling of Ni and Ni-Cu scraps with residues of NdFeB could be subject to further research, too, as both materials are valuable and critical materials.

In general, knowledge of current coating and fixation methods greatly improves the chances of efficient recycling while recognizing that new recycling technologies may be developed between the design of current magnet-containing products and their end-of-life. Adequate and standardized labeling [12] is one important factor in increasing the quality and yield of recycled NdFeB powder generated via the HPMS short-loop recycling route.

## Figures and Tables

**Figure 1 materials-17-02487-f001:**
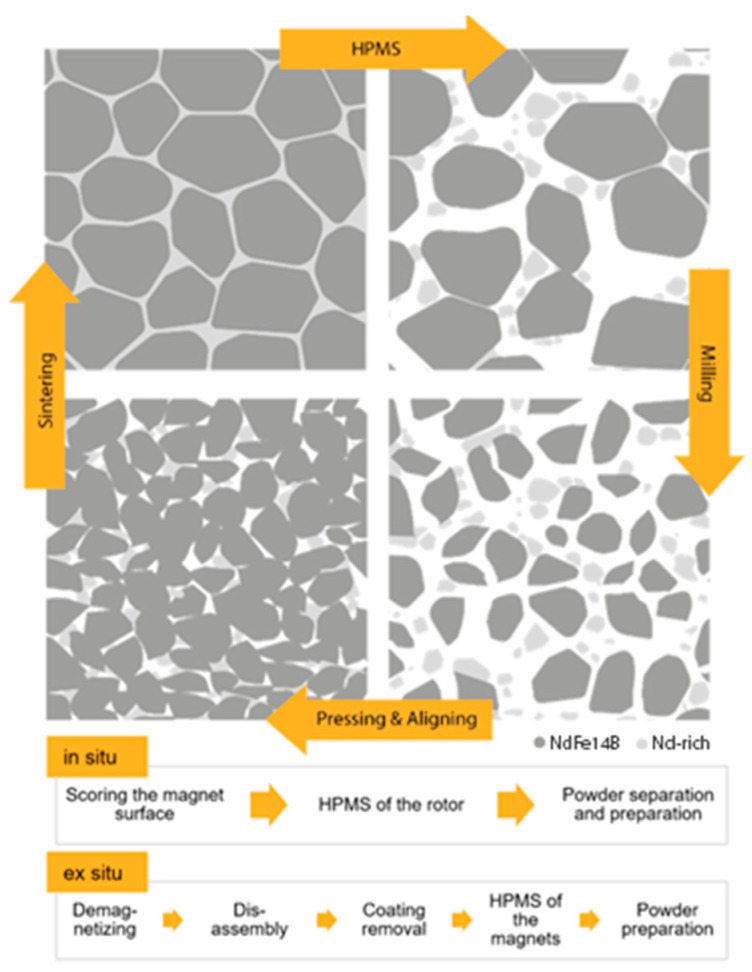
Circular lifecycle of NdFeB using HPMS and re-sintering, process comparison of in situ and ex situ treatment in HPMS seen below: in situ-treated magnets are exposed to hydrogen while still mounted and decrepitated to powder, which must then be removed and separated from coating residues; ex situ-treated magnets are demounted and their coatings are removed prior to treatment.

**Figure 2 materials-17-02487-f002:**
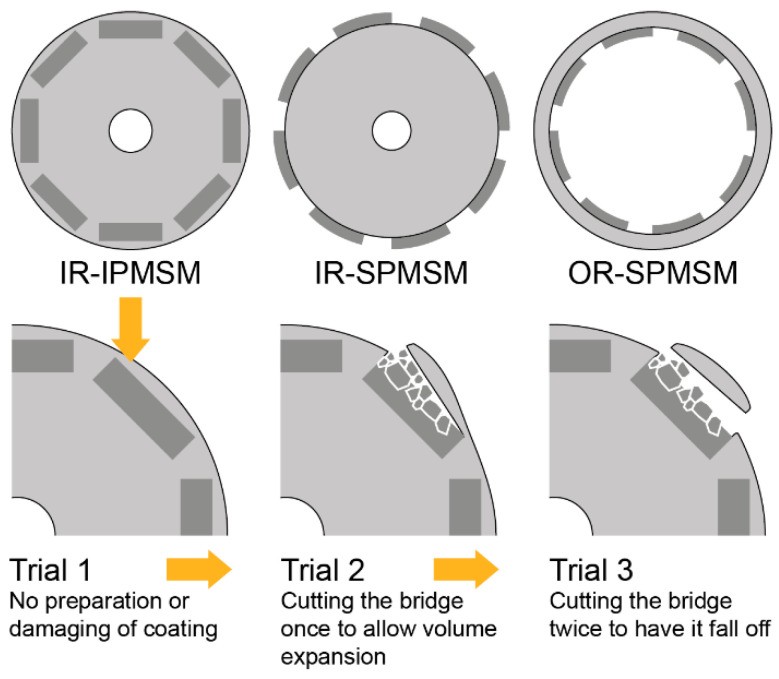
Tested rotor arrangements visualized: inner rotor (IR) arrangement of internal permanent magnet synchronous motors (IPMSMs) as well both IR and outer rotor (OR) arrangements of surface mounted permanent magnet synchronous motors (SPMSMs). The preparation methods for the in situ trials are visualized below.

**Figure 3 materials-17-02487-f003:**
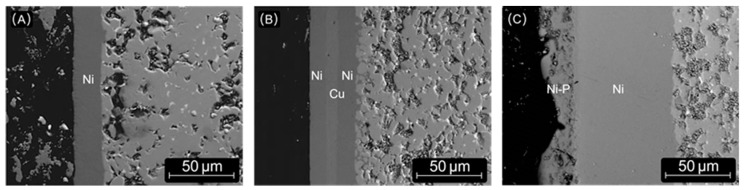
SEM-BSE images of single Ni (**A**) Ni/Cu/Ni (**B**) multilayer Ni with Ni-P (**C**) as observed in the samples used in this investigation. Samples shown in (**A**,**C**) are derived from mixed scrap samples 8 and (**B**) is derived from sample 6/7. Hitachi FlexSEM 1000II (Tokyo, Japan), W cathode, 15 kV, BSE-COMP, 200× magnification.

**Figure 4 materials-17-02487-f004:**
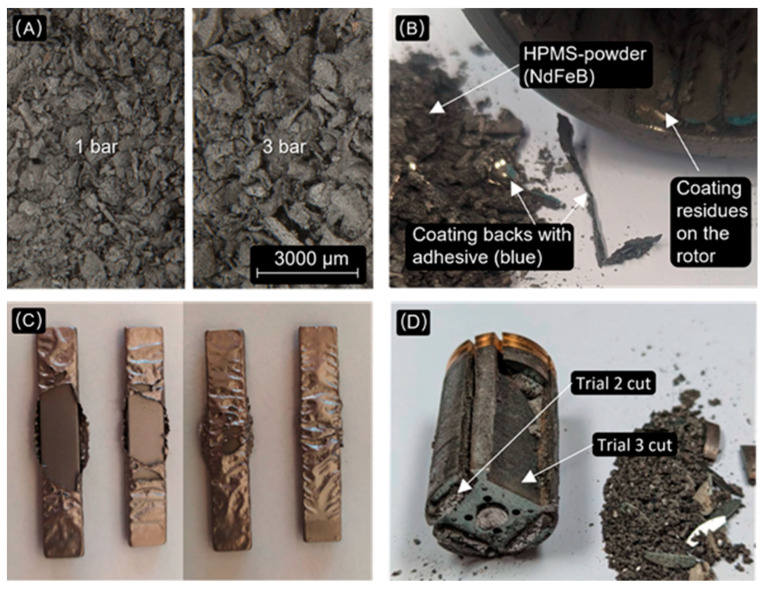
Comparison of the same type of magnets (mounted in sample 1) being treated at 3 bar and 1 bar for 1 h imaged by Leica M205 C (Wetzlar, Germany), 10× magnification (**A**), powder contaminated by coating backs and adhesive after in situ testing of sample 6 (**B**), failed sandblasting of sample 7 magnets (**C**) and results after cutting sample 3 once and twice, showing the decrepitated powder intermixed with rotor sheet parts next to the rotor (**D**). For sample details, see Table 1.

**Figure 5 materials-17-02487-f005:**
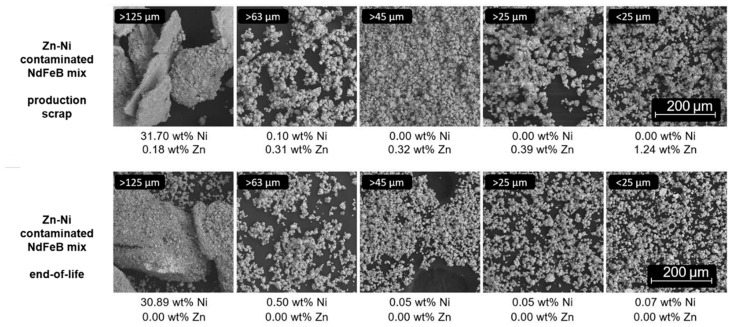
Ni and Zn content of sieved fractions determined by ICP-OES of low-corrosion production scrap (sample 7, sample 2, same manufacturer) and high-corrosion end-of-life scrap (sample 8)—the amounts of powder in the images are not indicative of the amounts found in each fraction. Hitachi FlexSEM 1000II, W cathode, 15 kV, BSE-COMP, 200× magnification.

**Table 1 materials-17-02487-t001:** Samples tested in the investigations. Supplied in a magnetized condition.

Rotor Sample	Motor Type	Magnet Mounting	Coating	Scrap Type
1 IR-IPMSM-Zn	Inner rotor	Internal, press-fit	Zn	Prod. scrap
2 IR-IPMSM-Zn	Inner rotor	Internal, press-fit	Zn	Prod. scrap
3 IR-IPMSM-Zn mix	Inner rotor	Internal, glued	Zn	End-of-life
4 IR-IPMSM-P	Inner rotor	Internal, press fit	Phosphate passivation	Prod. scrap
5 IR-SPMSM-ZnEp	Inner rotor	Surface mounted, glued	Zn-Epoxy	Prod. scrap
6 OR-SPMSM-NiCuNi	Outer rotor	Surface mounted, glued	Ni-Cu-Ni	Prod. scrap
**Disassembled magnet samples**		**Source**	**Magnet mounting**	**Scrap type**
7 Ni-Cu-Ni		Sample (6)	Demounted	Prod. scrap
8 Ni- and Ni-Cu-based mix		Hard disk drives	Demounted	End-of-life

## Data Availability

The data presented in this study are available on request from the corresponding author due to privacy reasons.

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
