# Peer review of "Processability and Separability of Commercial Anti-Corrosion Coatings Produced by In Situ Hydrogen-Processing of Magnetic Scrap (HPMS) Recycling of NdFeB"

_materials, 2024, doi:10.3390/ma17112487_

Round 1

Reviewer 1 Report

Comments and Suggestions for Authors

Major:
- Figure 1, there are three parts of this figure: the cycling, in situ, and ex situ. The figure caption needs to provide some context for each of these parts. At the moment, the reader might be confused by the lack of explanation for the in situ and ex situ parts of the figure. The main text also does not explain what ex situ is, relative to HPMS and in comparison to in situ.

-Section 2.2/2.3,  the description of the different types of coatings in 2.2 is appreciated, but this materials section should specifically state at its beginning that the following 4 types of coatings were studied in this specific study. At the moment, it looks like a small mini-review inside a primary research paper, which doesn’t make sense until 2.3 in Table 1 where the reader needs to see column 4 to make sense of the purpose of 2.2.

-Line 175, please describe which ICP-OES instrument was used, and the protocol for preparing samples for ICP-OES, the ICP-OES settings, and the reagents used (e.g., which nitric acid, tuning solution, etc.).

-Figure 4, do these magnets correspond to the samples listed in Table 1? Online Sample 6 and 7 are described. Are all the magnets here belonging to 6 and 7?

-Line 217-224, does this paragraph correspond to any data? Is there a figure that can be referenced?

-Figure 5, can the authors make it clear which of the samples in Table 1 do these magnets tested here correspond to. The caption should also state what kind of imaging was performed.

- Overall, the manuscript does not clearly explain the connection between the samples and the quantitative results. The pictures of the magnets do not correspond with Table 1’s labelling of the samples, and the exact source of the rotor samples is not provided, thus limiting any reproducibility studies outside of the authors’ research group. Moreover, the study relies too much on qualitative descriptions of photos. Figure 4 in particular should be replaced with quantitative data. For example, 4A is a picture of processed magnets (not clear which magnet?), but the source of light for the photo shines mostly on the 1bar sample, while the 3bar sample is dark. To the untrained eye, these look the same. For 4B, the authors are showing magnet scraps on a metal surface, so it is difficult to actually make any interpretation of what the reader is looking at besides some metal on a metal surface. For 4C, which magnets are 6 and 7? Figure 5 provides some quantitative data, but it is not referenced in the main text.

Minor:
-Line 52, please adjust the sentence to not use “i.e.” mid-sentence. You could remove “i.e. mounted in small rotors” and the sentence would still work.

-Line 60, remove the second “then”.

-Table 1, what is “Phos.”

-Figure 3, what do the acronyms stand for? Please include this in the captions.

-Please refer to the different parts of the papers as “Sections”, not “Chapters”.

- Line 188, what does IPMSM stand for?

-Figure 4, forgot to include label for panel 4D.

-Figure 4, in panel 4D, what is the milled material on the right?

Comments on the Quality of English Language

It's fine. Some parts are too colloquial. 

Reviewer 2 Report

Comments and Suggestions for Authors

In this manuscript, the authors proposed the recycling procedures for the NdFeB hard magnet components in the motor or HDD devices. The impact of various coating materials on the performance of HPMS recycling were discussed and tested. While the paper is quite clear, it could be further improved by the following suggestions and discussions.

1. As one of the major presented experiments, the sieving procedures conducted in this manuscript separate the coating and NdFeB materials. I wonder what the percentage of NdFeB can be sieved out after milling comparing to original magnet component, or how many NdFeB can be recycled after this process. Also how the mill process parameters affect the performance of separation, like mill duration, ball to powder ratio, etc.

Similar thing for the Zn-coating layer removal, can you estimate how much NdFeB can be recycled after sandblasting of Zn-based layer?

2. Authors discussed the influence of different coatings, and then briefly suggested the proper treatment method of each type. I am interested how to identify the different coating materials in practice for a recycling motor, just by observation, or some pre material examination, or have to find out the products information.

3. Talking about the recycling process, the coating materials might not be that crucial comparing to the raw earth magnet, but just wonder if we apply recycling also for the coating materials itself. For example, the sieved out large particles of Ni-Cu-Ni flakes.

Round 2

Reviewer 2 Report

Comments and Suggestions for Authors

I think authors have addressed all my comments thoroughly and have enrich the discussion of recycling efficiency and future research plan in the manuscript. I would suggest publish this manuscript in "Materials" as present form.